# Pragmatic selection of larval mosquito diets for insectary rearing of *Anopheles gambiae* and *Aedes aegypti*

**Mark Q. Benedict**[1]*, **Catherine M. Hunt**[1], **Michael G. Vella**[2], **Kasandra M. Gonzalez**[2], **Ellen M. Dotson**[1], **C. Matilda Collins**[3]

**1** Division of Parasitic Diseases and Malaria, Centers for Disease Control and Prevention, Entomology Branch, Atlanta, Georgia, United States of America, **2** Frontier Scientific Services, Newark, Delaware, United States of America, **3** Centre for Environmental Policy, Imperial College London, London, United Kingdom

* mbenedict@cdc.gov

**Data Availability Statement:** Data underlying the study are available on the OSF repository (https://osf.io/zc6dr/).

## Abstract

Larval mosquitoes are aquatic omnivorous scavengers which scrape food from submerged surfaces and collect suspended food particles with their mouth brushes. The composition of diets that have been used in insectaries varies widely though necessarily provides sufficient nutrition to allow colonies to be maintained. Issues such as cost, availability and experience influence which diet is selected. One component of larval diets, essential fatty acids, appears to be necessary for normal flight though deficiencies may not be evident in laboratory cages and are likely more important when mosquitoes are reared for release into the field in e.g. mark-release-recapture and genetic control activities. In this study, four diets were compared for rearing *Anopheles gambiae* and *Aedes aegypti*, all of which provide these essential fatty acids. Two diets were custom formulations specifically designed for mosquitoes (Damiens) and two were commercially available fish foods: Doctors Foster and Smith Koi Staple Diet and TetraMin Plus Flakes. Development rate, survival, dry weight and adult longevity of mosquitoes reared with these four diets were measured. The method of presentation of one diet, Koi pellets, was additionally fed in two forms, pellets or a slurry, to determine any effect of food presentation on survival and development rate. While various criteria might be selected to choose 'the best' food, the readily-available Koi pellets resulted in development rates and adult longevity equal to the other diets, high survival to the adult stage and, additionally, this is available at low cost.

## Introduction

Larval mosquitoes are omnivorous opportunistic aquatic feeders which collect and swallow small particles, can chew larger particles and can scrape food off of submerged surfaces [1]. Laboratory culture of mosquitoes seldom attempts to replicate natural diets but usually consists of a readily available material that experience has proven to allow consistent rearing. These generally fall into two classes: simple mixtures of ingredients such as yeast and liver powder that are formulated by the users or commercial formulations of complex composition including foods such as fish-food flakes [2] or pellets [3], hog supplement [4], cereals and less

**Funding:** MQB, EMD, and CMT are paid employees of Target Malaria, a project that receives core funding from the Bill & Melinda Gates Foundation and from the Open Philanthropy Project Fund, an advised fund of Silicon Valley Community Foundation to the Target Malaria project, PI Austin Burt. MV and KG are paid employees of Frontier Scientific Services which supplied the formulation to the CDC with the understanding that the experimental design and diet comparisons would not be influenced by the potential for commercialization. Frontier kindly formulated the diet and provided it without charge for these comparisons. MV and KG reviewed the manuscript for style and accuracy. The following reagents were obtained through the NIH Biodefense and Emerging Infections Research Resources Repository, NIAID, NIH: An. gambiae, strain 'G3' (MRA-112) and Ae. aegypti 'New Orleans' strain (NR-49160). The funders had no role in study design, data collection and analysis, decision to publish, or preparation of the manuscript. The specific roles of these authors are articulated in the 'author contributions' section.

**Competing interests:** The authors have read the journal's policy and the authors of this manuscript have the following competing interests: MQB, EMD, and CMT are paid employees of Target Malaria. MV and KG are paid employees of Frontier Scientific Services which supplied the formulation to the CDC. This does not alter our adherence to PLOS ONE policies on sharing data and materials.

commonly, maize pollen and algae [5]. Many unusual ingredients such as guinea pig feces and hay infusion are cited by Gerberg [6]. The diversity of 'successful' larval foods demonstrates that for many purposes, there are numerous choices.

Commercially manufactured diets provide the advantages that the researcher does not need to formulate the diet, can rely on the quality control measures employed by the manufacturer and often, ready availability. Some disadvantages of complex commercial foods are that the researcher has no control over the specific components of the diet which may change without notice and obtaining the same diet locally in different countries may be difficult.

Considerations for choosing a food are simple: It must promote the routine rearing of good quality mosquitoes (however that is defined), be readily available, consistent in quality and preferably inexpensive. A less apparent and seldom considered advantage is to choose a diet that numerous laboratories can use to give some assurance of comparable results. If flight testing, mating assays or release activities will be performed, it is necessary to provide essential fatty acids [7].

This study determined, among four candidate larval diets for two frequently-reared disease vector mosquitos, *Anopheles gambiae* Giles (Diptera: Culicidae) and *Aedes aegypti* Linnaeus (Diptera: Culicidae), which diet and feeding level resulted in the optimal performance for several important life history traits such as immature growth rate, survival, size and adult longevity. One of these diets, TetraMin flakes, is widely used for both *Anopheles* and *Aedes spp*. We make a recommendation for selection among these diets which also considers cost and availability.

## Materials and methods

### Diet preparation

Four diets were prepared for comparisons; two of these were custom formulations of a diet specifically designed for mosquitoes developed by Damiens et al. [8]. This diet consists of a 2:2:1 ratio (by weight) of bovine liver powder, tuna meal and Vanderzant vitamin mix. One formulation was prepared at CDC in Atlanta, GA using 'Now' brand liver powder (Bloomingdale, IL USA), tuna meal (AA Baits, Rock Ferry, Birkenhead, UK) and Vanderzant vitamin mix (Bio-Serv, Flemington, NJ, USA). Large particles were removed from the tuna meal and liver powder using a (600 μ) standard sieve. Clumps of vitamin mix were broken up manually but no further sieving was done because the mix is soluble and the particle size allowed even mixing.

The other formulation of the Damiens diet was prepared by Frontier Scientific Services (Newark, DE USA) using defatted, desiccated liver powder (product no. 1320; Frontier Scientific Services), Vanderzant vitamin mix (product no. F8045, Frontier Scientific Services) and the same lot of tuna meal as was used at CDC. In order to ensure particle size was small enough for consumption by developing larvae, a milling and screening procedure was employed. A significant source of oversized particulates was the tuna meal. Most large particles were identified as scale and bone remnants from the manufacturers processing of the meal. The tuna meal was processed in a top-feeding hammer mill (The Fitzpatrick Co., Toronto, Canada) with a 60–80 (177 μ) mesh particle excluding screen. To ensure particles were milled to specification without complete exclusion of meal components, the material was passed through the hammer-mill twice. The milled tuna meal was then mixed with the remaining ingredients in a bench-top 'Kitchen Aid' bread mixer for 20 minutes. After mixing was complete, the final diet was hand sifted through a 60 mesh (177 μ) screen to eliminate any remaining oversized particulates.

The other two diets were commercially available fish foods: Doctors Foster and Smith Koi Staple Diet (Rhinelander, WI USA) and TetraMin Plus Flakes (Tetra GmbH, Melle, Germany). For fair comparison, both fish foods were ground to a similar size to the custom diets. Koi

pellets were ground in a Miracle Model MR-300 Electric Grain and Flour Mill (Danbury, CT USA) followed by sieving in a 600 μ standard sieve and saving the particles that passed. The TetraMin was ground in a Black and Decker 'SmartGrind' coffee grinder (Beachwood, OH USA) after which it easily passed through a 600 μ sieve. These diet types will be identified as CDC, Frontier, Koi, and TetraMin respectively.

The ground diets were mixed at 0.4, 0.8, 1.6 and 3.2% w/v in type II water and stored in ca. 13 ml aliquots and frozen at -20˚C where they remained until being thawed in warm water immediately before feedings. When 4 ml of the slurry was fed, these concentrations result in feeding rates (levels) of 8, 16, 32 and 64 mg diet / dish / day. Hereafter we will usually refer to the levels simply as *e.g.* 32 mg.

## Mosquitoes

*Anopheles gambiae* mosquitoes were the 'G3' strain (MRA-112) obtained from the Malaria Research and Reference Resource Center (MR4, BEI Resources, Manassas VA USA). *Aedes aegypti* were the 'New Orleans' strain (NR-49160), also obtained from the MR4 and were in the F16-F18 generations during experiments. A standard rearing water was made of 0.3 g of pond salts (API, McLean, VA USA) per liter of type II purified water. *Anopheles gambiae* eggs were collected, held overnight on damp filter paper and placed in trays on the day of hatching. *Aedes aegypti* eggs were dried for embryonation for 4 days after collection before storage in plastic bags in a plastic box. Within 1 month, they were hatched by placing egg papers in water under vacuum for 30 min. Hatching embryos of both species were placed in 500 ml of water containing 5 intact Koi pellets for one day before counting 80 larvae into 150 ml polystyrene Petri dishes (Item no. Z717231, Sigma-Aldrich, St. Louis, MO USA).

## Trial design

An orthogonal design was used; three dishes (replicates) for each of the four diets at all four levels were established for both species (Fig 1). For these mosquitoes, it is not possible to determine sex at the first larval instar and it was assumed that random aliquots would deliver a representative sex ratio. Before counting larvae into Petri dishes, the empty dishes were weighed to a tenth of a gram on a triple-beam balance (700/800 Series, Ohaus, Parsippany, NJ USA) and labeled with their weight. On the day after hatching, 80 larvae were counted into the dishes and rearing water was added until the net weight was 96 g. Then 4 ml of food was added for an approximate total volume of 100 ml. The concentrations were selected to bracket a range shown to allow maximal survival and development rate with *Anopheles arabiensis* [9]. Additional diet was added on alternate days, prior to which the dishes were weighed and water was removed (ca. 2–3 ml), to return the net weight to 96 g before 4 ml of diet slurry was added to maintain an approximate total volume of 100 ml. Mosquitoes were reared in an environmental room set at 27˚C and 70% relative humidity with a 12:12 light:dark cycle and 30 minutes of dawn and dusk.

In this main experiment, in which all diets and levels were tested, the dishes were inspected daily and pupae were counted and collected in the morning daily beginning on the first day they were observed. After collection, their sex was determined and the pupae were then placed in individual plastic tubes for eclosion. Tubes were checked for adults daily with up to five randomly selected adults from each day of eclosion and sex being killed for dry weight measurements. Immature stage trials were generally terminated when there were no more larvae present except as noted for the 8 mg diet level with *An. gambiae* where observations of larval duration were terminated based on a pragmatic decision on days 12 and 14 (Table 1).

80 L1 / dish (x 3)

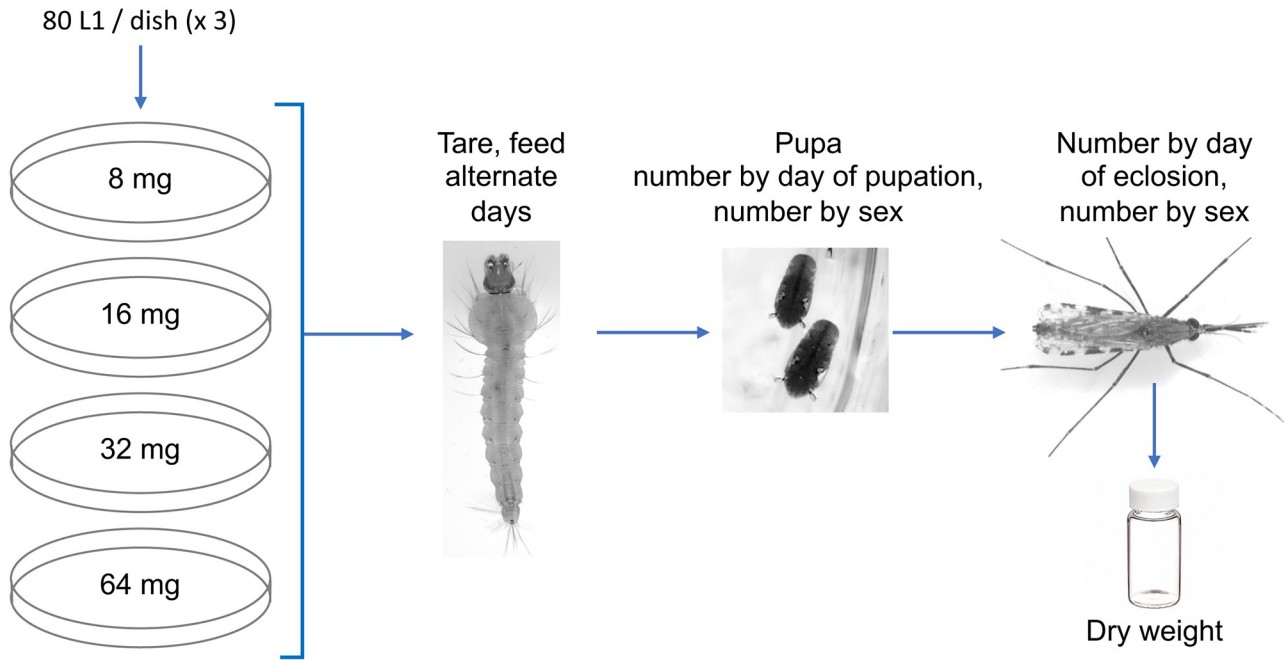

**Fig 1. Experimental trial design.** For each diet type tested, 12 dishes were observed, three at each diet type and level. When pupae formed, they were collected daily, their sex determined and transferred to tubes for eclosion. Adults were removed daily, dried overnight and weighed.

*Anopheles gambiae* and *Ae. aegypti* differ in many characteristics including body size and rearing tractability. Because of this, the two species have been analyzed separately. There are also known differences in outcomes by sex within species and, where appropriate, parameter estimates for each sex are calculated independently. Statistical analyses were performed using R version 3.5.1 "Feather Spray" [10]

**Inter-trial comparison of water temperature.** Due to logistical limitations, it was not possible to perform all experiments concurrently. As a result, five sequential trials in the same chambers contributed to the experiment overall. The critical variable of water temperature was measured in three arbitrary dishes every two or three days in the morning using a Sper

**Table 1. The number of *An. gambiae* immatures discarded at the end of the trial at the lowest diet level, 8mg.**

| Diet type | Dish | Day | Discarded |
|---|---|---|---|
| CDC | A | 14 | 47 |
| | B | | 28 |
| | C | | 48 |
| Frontier | A | 14 | 13 |
| | B | | 7 |
| | C | | 10 |
| Koi | A | 12 | 51 |
| | B | | 52 |
| | C | | 47 |
| TetraMin | A | 12 | 63 |
| | B | | 66 |
| | C | | 57 |

Scientific Model 800005 thermocouple thermometer (Scottsdale, AZ USA) equipped with a K type probe and overall means were compared using Analysis of Variance.

**Sex ratio.**　The sex of pupae arising from each treatment combination was observed and the ratio estimated. Chi-square tests were used to determine whether the male:female ratio varied with treatment or species.

**Survival to eclosion.**　To determine the effect of the different diets and levels fed on the number of adults that eclosed, Poisson-family generalized linear models (GLMs) using the diet level, diet type and their interaction were fit to the data. Model simplification by deletion tests used F tests to estimate influential effects as appropriate to the over-dispersion of these count data.

**Proportion of pupae eclosing.**　The data that were analyzed resulted from the counts of the number of pupae that formed and eclosion data. A weighted response variable that bound the number of pupae eclosing and the number of pupae that did not was created. Binomial-family GLMs using the dose, food type (both categorical, four levels) and their interactions were fit to this data. Model simplification by deletion used F tests to estimate important effects as appropriate to the over-dispersion evident in the weighted proportion data.

**Larva developmental rate.**　The number of days taken to complete larval development to pupation was analyzed to determine effects on development rates. As the number of days to eclosion was an integer value, chi-squared tests were used to estimate the influence on this time of interactions between diet type and the amount of food provided as well as these as single effects. The relative contribution of each factor is then reflected in the test statistic values.

**Adult longevity estimation.**　The 32 mg diet level was chosen for assessing the influence of diet type on adult longevity based on observed rapid development rate and high survival across all diet types reported in the Results section. Pupae arising from these dishes were placed in aluminum-frame cages [11] which were covered with one or two layers (in the case of *Ae. aegypti)* of gauze and provided sugar water (10% w/v food grade sucrose, 0.1% w/v methylparaben in type II water) which was changed weekly. There were three cages for each diet, each associated with a different larval replicate dish. All longevity measures were made concurrently. Mortality was usually checked daily, though occasionally it was not observed on Saturdays. Kaplan-Meier objects were created as response variables for the survival analyses and a Cox proportional hazards model was used to identify effects of diet type on survival for each species and sex.

**Measures of dry weight.**　Dry weight of adults was determined for all diet types and levels. After eclosion, adults were transferred to glass scintillation vials, killed by freezing at -20˚C and dried in a drying oven at 60˚C overnight after which they were removed and the caps sealed until weighing. For each diet/level combination, up to five individuals of each sex from each day of eclosion were weighed using a Sartorius SuperMicro S4 balance (Bohemia, NY USA) when that number was available. Weights are reported in micrograms.

The dry weight of mosquitos was a continuous response variable. As previously, diet type, level and mosquito sex were all considered as categorical factors. All main effects and interactions were tested by deletion from the maximal model. Effects that were either non-significant or accounted for less than 1% of the variation in the data were excluded.

**The influence of pellet vs. slurry.**　One food type, the Koi pellets, was used to estimate any influence of the form of presentation and thus whether it is necessary to grind the food. Koi pellets were weighed on the SuperMicro balance and the average weight and standard deviation of pellets calculated; 52.0 mg (n = 14, StDev 8.65). Two pellets (equivalent to 52 mg/dish/day) were fed on alternate days in parallel with the day the 32 mg slurry was given. Larval survival (the number of larvae reaching pupation) and larval duration (the number of days to pupation) were used as the measures for this comparison. Pupae were collected daily in the

morning and their sex determined. All dishes were new and there were three tests of each food form for both *Ae. aegypti* and *An. gambiae*.

## Results

### Inter-trial comparability of water temperature

The temperature was consistent among all trials of *Ae. aegypti* (F = 1.03, d.f. = 2,72 p = 0.36). The average water temperature was 26.9˚C (n = 75, StDev 0.45). The average temperature of all *An. gambiae* trials was 27.0˚C (n = 45, StDev 0.33) but there was a slight, but significant, variation in temperature between the trials (F = 7.51, d.f. = 1.43, p<0.01); a trial during which Koi and TetraMin were being tested was on average 0.25 +/- 0.1˚C lower than one in which CDC and Frontier were being tested. This effect is, however, largely driven by a single day, day 7, in the CDC-Frontier trial which was warmer than other days (t = 2.49, d.f. = 14, p<0.05).

### Sex ratio

The proportion of *Ae. aegypti* pupae was observed to be consistently male-biased (0.59 (95% CI: 0.57–0.60) relative to an assumption of equal proportions of males and females. Because all pupae that formed were collected, this bias could not be due to male pupae forming earlier than females. Nor can it be due to differential mortality. Assuming all mortality in the trials consisted of females, the ratio would still be male-biased (0.54). In contrast, the overall ratio of male pupae in *An. gambiae* (0.47, 95%CI: 0.43–0.51) did not vary from equal proportions of either sex.

### Survival from hatch to eclosion

In the *An. gambiae* 8 mg experiments, pupa formation was so prolonged and low that many larvae and pupae were discarded on day 12 or 14 of larval development, so interpretation of the results should take this into account (Table 1). Discarded pupae were not included in the analysis of likelihood to eclose.

The responses of *Ae. aegypti* and *An. gambiae* to different diets and levels shared similarities in pattern but had marked differences in absolute level (Fig 2). *Aedes aegypti* eclosion varied less as a function of diet type and level and achieved higher numbers than did *An. gambiae* (for model null deviance see Table 2). There was an interaction between diet type and level on the number of *Ae. aegypti* males and females; this was largely driven by the two commercial foods, Koi and TetraMin, having higher numbers at the lowest diet level than did the CDC or Frontier diets.

The *An. gambiae* pattern was the same for both males and females, and there was a slight interaction between diet type and the food level (this explained ca. 10% of the deviance in the data for each sex) largely driven by the very low numbers forming from TetraMin at both the lowest and highest dose. It must, however, be remembered that individuals were discarded earlier in the experiment due to slow development at the lowest diet level for the TetraMin diet (Table 1). The magnitude of the effect of diet level was much greater for *An. gambiae* than for *Ae. aegypti* and, overall, few *An. gambiae* eclosed at both the lowest and highest diet level.

**Proportion of pupae eclosing.** We anticipated that the likelihood of pupae that had formed then successfully eclosing might be affected by the food type or level. For both sexes of *Ae. aegypti* there was an interaction between diet type and level on the number of pupae eclosing to adults (Table 3); this was largely driven by poor eclosion at low levels of the Frontier diet (Fig 3). In all other cases, if the larvae reached the pupa stage, they were highly likely to become an adult.

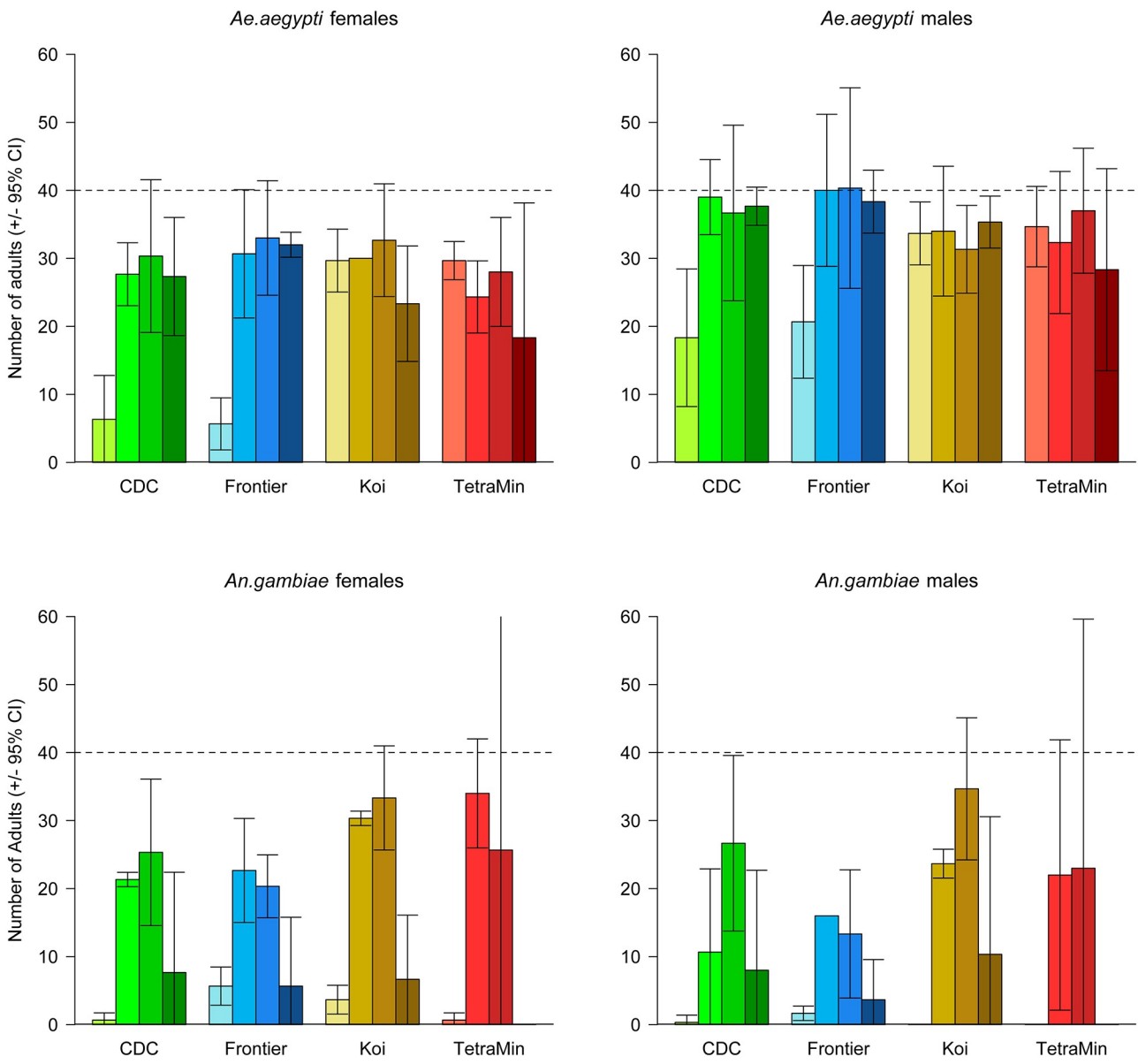

**Fig 2. The number of *Ae. aegypti* and *An. gambiae* female and male adults observed by diet type and level.** The dashed horizontal line indicates the expected number of females and males assuming a 1:1 sex ratio and full survival. Error bars are the 95% CI of the mean. Darkening shades of color represent the increasing diet levels of 8, 16, 32 and 64 mg.

For *An. gambiae*, the pattern was similar for both sexes. Pupae that resulted from feeding on the Koi diet were most likely to eclose regardless of diet level (Fig 3), though generally the eclosion rate was highest at the intermediate diet levels than it was at either the highest or lowest levels (Table 3).

The magnitude of the treatment effects was much greater for *An. gambiae* than for *Ae. aegypti* (Table 3) with, overall, the *An. gambiae* being more sensitive to the type and level of the diet provided (Fig 3). The response of *Ae. aegypti* was more nuanced with only the interaction between diet and level being significant.

**Table 2. Statistical summary of the influences on the number of male and female adults formed with the proportion of the deviance explained (in parentheses) given as an indicator of effect size (significant effects indicated in bold).**

| | *Ae. aegypti* | | *An. gambiae* | |
| | Males | Females | Males | Females |
|---|---|---|---|---|
| Model null deviance (47 d.f.) | 86.77 | 201.09 | 689.35 | 669.89 |
| Diet:Level | **F = 5.11, d.f. = 9,32, p<0.001 (0.43)** | **F = 12.53, d.f. = 9,32 p<0.001 (0.55)** | **F = 2.30, d.f. = 9,32, p = 0.04 (0.10)** | **F = 2.82, d.f. = 9,32, p = 0.015 (0.10)** |
| Diet | F = 0.18, d.f. = 3,41, p = 0.91 (0.015) | F = 0.92, d.f. = 3,41, p = 0.44 (0.04) | F = 2.67, d.f. = 3,41, p = 0.06 (0.05) | F = 1.06, d.f. = 3,41, p = 0.38 (0.02) |
| Level | **F = 5.69, d.f. = 3,44, p = 0.002 (0.26)** | **F = 5.15, d.f. = 3,44, p = 0.004 (0.28)** | **F = 28.70, d.f. = 3,44, p<0.001 (0.64)** | **F = 42.92, d.f. = 3,44, p<0.001 (0.70)** |

## Immature development

For both species and sexes the pattern is similar. The time taken to complete the larval stage is an interaction of both the food type and the level (p<0.05 in all cases; Table 4, Fig 4) except for *An. gambiae* females for which there was not sufficient evidence of an interaction to be conclusive. The effect of diet level is consistently much greater than variation observed among diet types.

*Anopheles gambiae* is the more sensitive species to diet level, but the estimates of time taken were based on many fewer measures than were possible for *Ae. aegypti* because of the low number of pupae at low and high doses. No estimates of larval duration were possible for *An. gambiae* in four of the combinations as none developed successfully. Generally, development times for *An. gambiae* were more consistent with Frontier, though there were few developing at low and high doses.

## Longevity of adults from 32 mg diet level larvae

The cage from which the individual mosquitoes came was included in each model to account for any cage effects; in no case were these identified to account for significant variation in the data (p>0.05 in all cases). Overall, the median adult lifespan of *Ae. aegypti* males and females was similar (Table 5). For females, there was no identifiable variation in longevity as a function of diet type ($\chi^2 = 4.45$, d.f. = 3, p = 0.22). There was variation in male longevity but CDC and Koi led to longer-lived males ($\chi^2 = 12.20$, d.f. = 3, p = 0.007).

*Anopheles gambiae* females lived consistently longer than males (Table 5). For females, diet type affected longevity with CDC and Koi leading to longer life ($\chi^2 = 9.87$, d.f. = 3, p = 0.02). There was greater variation in male longevity but no diet-related variation was identified ($\chi^2 = 5.80$, d.f. = 3, p = 0.12).

**Table 3. Model summary statistics estimating the influence of diet type and level on the number of male and female pupae eclosing to adults with the effect size (proportion of the deviance explained) indicated in parentheses (significant effects in bold).**

| | *Ae. aegypti* | | *An. gambiae* | |
| | Males | Females | Males | Females |
|---|---|---|---|---|
| Model null deviance | 165.10 | 155.63 | 207.64 | 96.12 |
| Diet:Level | **F = 3.33, d.f. = 9,32, p = 0.006 (0.35)** | **F = 3.19, d.f. = 9,32, p = 0.007 (0.33)** | F = 1.76, d.f. = 8,25, p = 0.14 (0.14) | F = 1.15, d.f. = 8,26, p = 0.37 (0.14) |
| Diet | F = 1.14, d.f. = 3,41, p = 0.35 (0.07) | F = 1.43, d.f. = 3,41, p = 0.25 (0.08) | **F = 4.31, d.f. = 3,33, p = 0.011 (0.16)** | **F = 4.53, d.f. = 3,34, p = 0.009 (0.19)** |
| Level | F = 2.24, d.f. = 3,41, p = 0.10, (0.13) | F = 2.51, d.f = 3,41, p = 0.07 (0.14) | **F = 10.10, d.f. = 3,33, p<0.001 (0.36)** | **F = 9.03, d.f. = 3,35, p<0.001 (0.21)** |

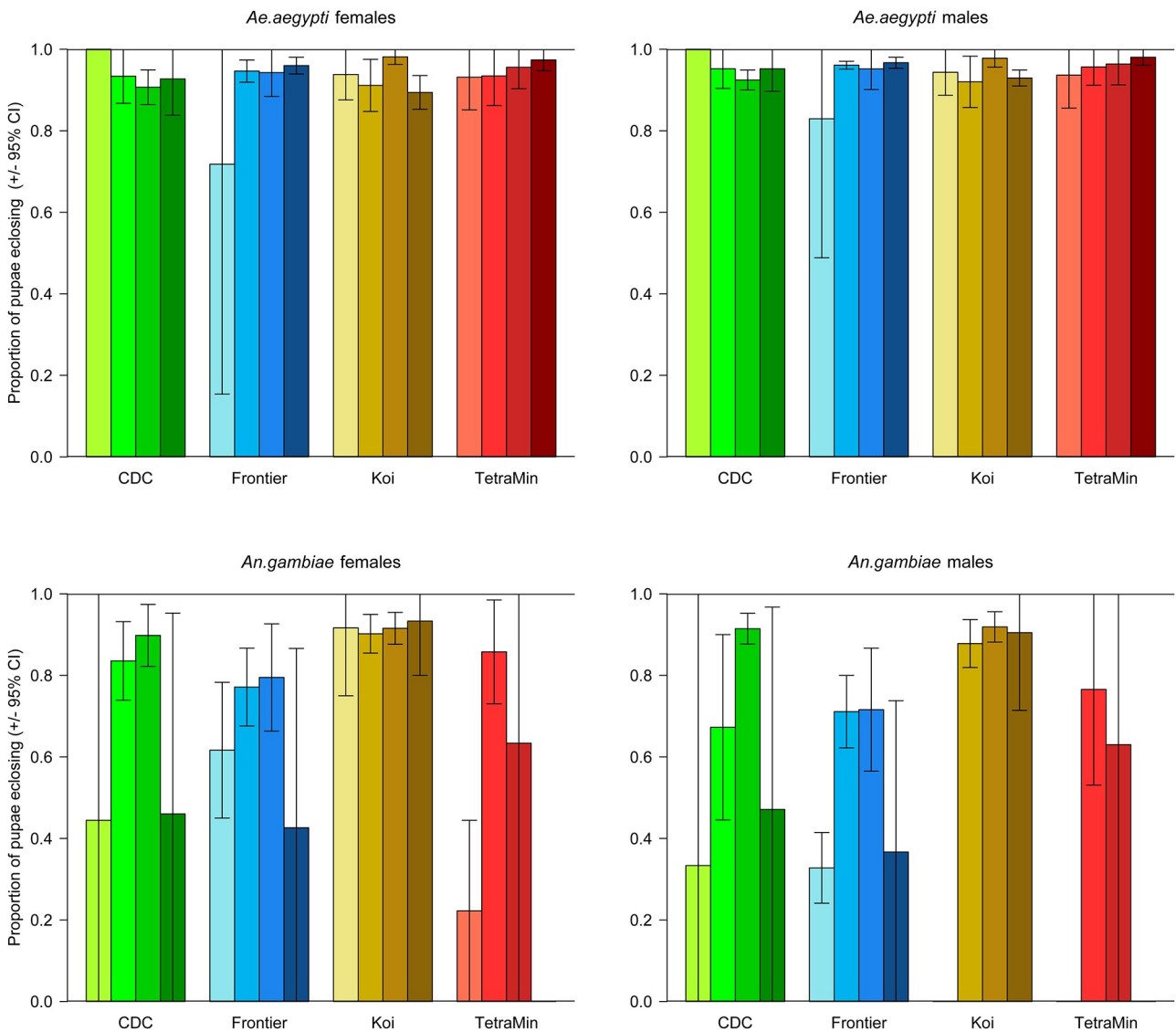

**Fig 3. Eclosion of pupae that formed by diet type and level.** Error bars represent the 95% CI. Darkening shades of color represent the increasing diet levels of 8, 16, 32 and 64 mg.

**Table 4. Development rate statistics.**

| | Ae. aegypti | | An. gambiae | |
|---|---|---|---|---|
| | **Females** | **Males** | **Females** | **Males** |
| Interaction Diet type:Level | $\chi^2 = 37.46$, df = 9, p < 0.001 | $\chi^2 = 26.31$, df = 9, p < 0.002 | $\chi^2 = 19.28$, df = 8, p < 0.05 | $\chi^2 = 19.19$, df = 6, p < 0.01 |
| Diet type | $\chi^2 = 12.77$, df = 3, p< 0.01 | $\chi^2 = 25.63$, df = 3, p<0.001 | $\chi^2 = 17.15$, df = 3, p < 0.001 | $\chi^2 = 23.35$, df = 3, p < 0.001 |
| Level | $\chi^2 = 150.61$, df = 3, p< 0.001 | $\chi^2 = 180.26$, df = 3, p < 0.001 | $\chi^2 = 77.55$, df = 3, p < 0.001 | $\chi^2 = 54.76$, df = 3, p < 0.001 |

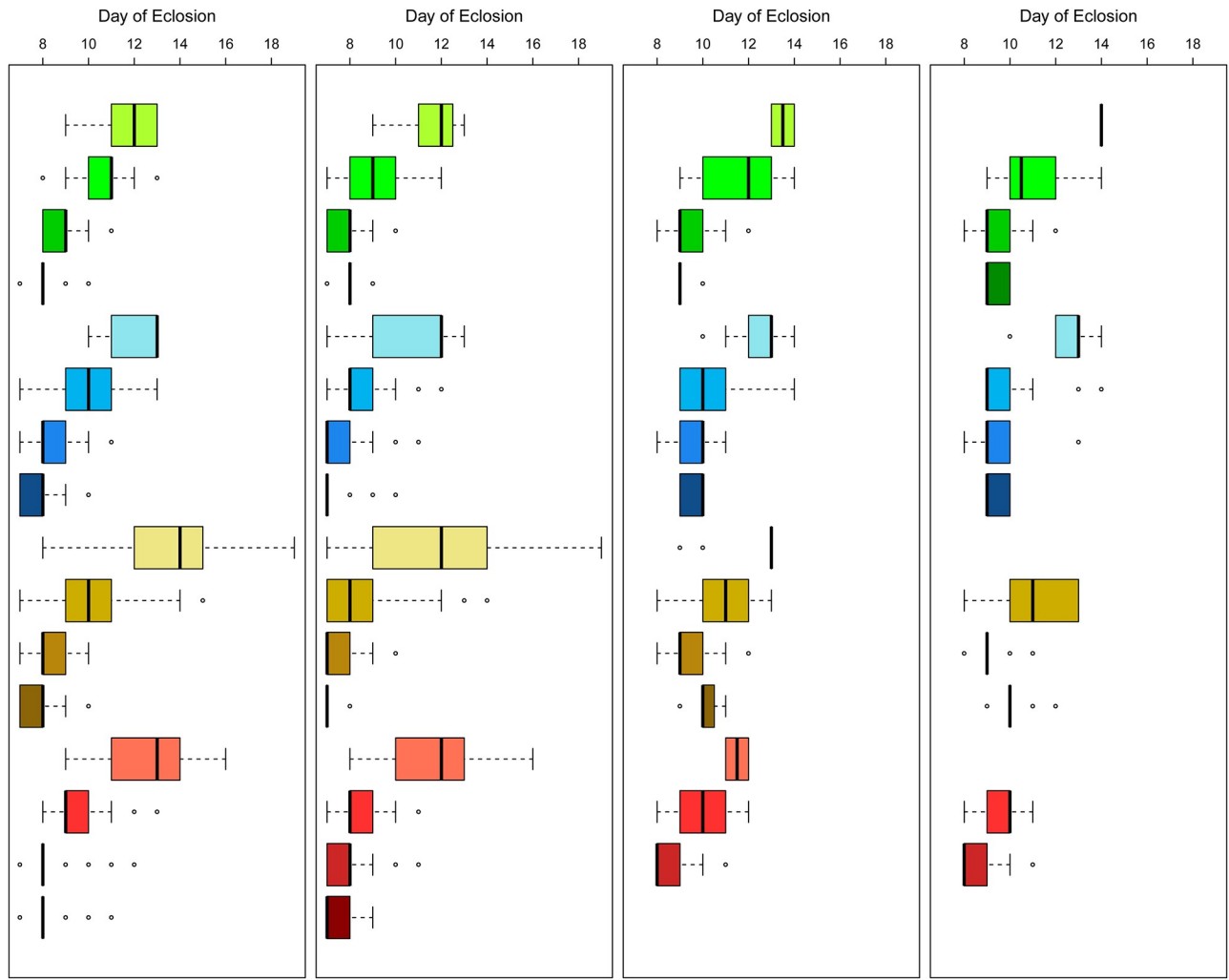

**Fig 4. Day of eclosion.** The panels are, left to right, *Ae. aegypti* females, males, *An. gambiae* females, males. The dark line indicates the median day of eclosion for each condition. The boxes contain the two central quartiles and the whiskers contain the outer quartiles unless outliers are present. These are indicated by points. Darker shades of color indicate increasing diet levels, 8, 16, 32 and 64 mg; CDC, green; Frontier, blue; Koi, brown and TetraMin, red.

## Dry weight

*Aedes aegypti* **dry weight.**    A total of 787 *Ae. aegypti* females and 880 males were weighed. *Aedes aegypti* males weigh less than females (p<0.001)–slightly more than half as much at any specific diet level. Across all diets, the ratio of male to female weight varied only slightly ranging from 0.56–0.58:1. Males were less responsive to increasing food quantity than females (Table 6, Fig 5). Though there was a small effect of diet type, the greatest effect for both sexes was that of diet level, which accounted for almost 50 times the variation than that found between diet types.

*Anopheles gambiae* **dry weight.**    A total of 208 *An. gambiae* females and 189 males were weighed. *Anopheles gambiae* males are lighter than females (p<0.001), but there was no difference in the way that they respond to the feeding regimes (all interaction terms >0.05). Feeding level had the strongest effect on adult weight, though food type was slightly influential; the mosquitoes responded differently to food level as a function of diet. The highest level of

**Table 5. Adult longevity.**

| | | *Ae. aegypti* | | | |
|---|---|---|---|---|---|
| | | Female lifespan (days) | | Male lifespan (days) | |
| Diet | n | Median (95% CI) | n | Median (95% CI) | |
| CDC | 55 | 51 (46–67) | 80 | 59 (56–67) | |
| Frontier | 52 | 49 (35–67) | 90 | 54 (51–61) | |
| Koi | 58 | 57 (53–63) | 82 | 60 (57–63) | |
| TetraMin | 69 | 50 (31–68) | 80 | 49 (44–53) | |
| | | *An. gambiae* | | | |
| | | Female lifespan (days) | | Males lifespan (days) | |
| Diet | n | Median (95% CI) | n | Median (95% CI) | |
| CDC | 47 | 37 (25–39) | 55 | 21 (14–26) | |
| Frontier | 57 | 32 (29–37) | 62 | 29 (27–32) | |
| Koi | 53 | 37 (37–37) | 48 | 20 (15–30) | |
| TetraMin | 66 | 30 (26–37) | 56 | 24 (18–29) | |

Frontier led to smaller mosquitoes, which was not the case for other diets. TetraMin gave low survival at highest and lowest doses and evaluations of adult mass were not possible there (Table 7, Fig 6).

## Food presentation: The influence of pellet vs. slurry on larval survival and development rate

The form in which Koi diet was fed had no effect on the number of pupae that formed in either species (Table 8).

Neither did the form affect the development rate of male larvae from hatch to pupation of either species (Table 8). However, the development of female larvae fed pellets delayed pupation by a day (median values: *Ae. aegypti* 7:6, *An. gambiae* 9:8 pellet vs. slurry respectively).

## Discussion

In this diet comparison, a range of diets fed at rates ranging from very low to high was compared. This experimental design was chosen to reduce the likelihood that the variation in the proportion of any particular component of diet (protein, fat or carbohydrates) might result in outcomes that do not represent the most favorable levels of diet fed. Because the ratios of protein, carbohydrates and fats differ among diets, a wide-level design is agnostic regarding which is most important for the outcomes tested. This approach is in contrast to Linenberg [3] in

**Table 6. *Aedes aegypti* weight statistics with significant effects shown in bold.**

| | F | d.f. | p | $R^2$ |
|---|---|---|---|---|
| **Full model** | **86.87** | **69,1597** | **<0.001** | **0.79** |
| Sex:Diet:Level | | | 0.52 | 0.00 |
| Sex:Diet | | | 0.14 | 0.00 |
| Level:Diet | | | 0.12 | 0.00 |
| **Minimal adequate model** | **88.88** | **10,1656** | **<0.001** | **0.78** |
| **Sex:Level (interaction)** | **75.94** | **3,1656** | **<0.001** | **0.02** |
| **Diet (factor)** | **21.96** | **3,1656** | **<0.001** | **0.01** |
| **Sex (factor)** | **1770.50** | **1,1659** | **<0.001** | **0.30** |
| **Feeding Level (factor)** | **909.51** | **1,1659** | **<0.001** | **0.45** |

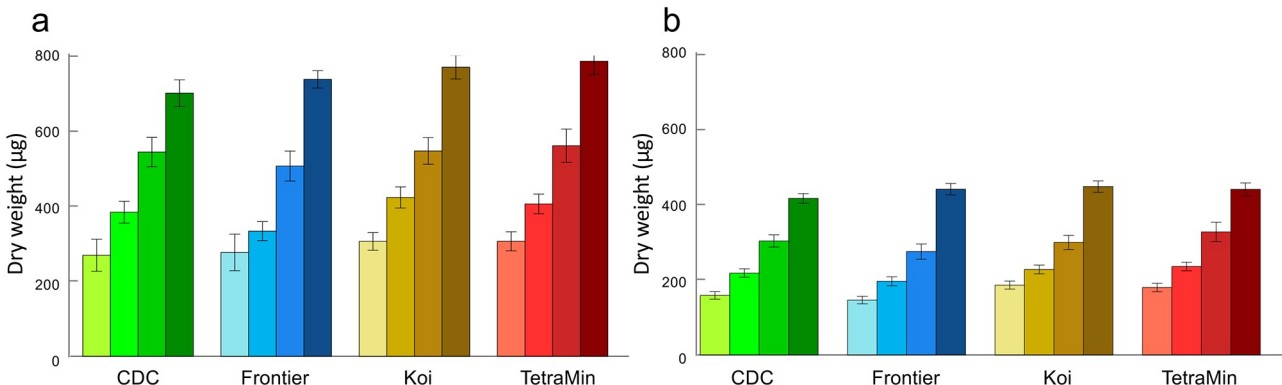

**Fig 5. Adult dry weights.** (a) *Aedes aegypti* females and (b) males. Darker shades of color indicate increasing diet levels of 8, 16, 32 and 64 mg. Error bars are 95% confidence intervals of the mean.

which the combined weight of fat and protein—to the exclusion of carbohydrates—was used to determine the amounts of diet provided to larvae for comparisons.

The reputation of *Ae. aegypti* as a robust and physiologically plastic laboratory model for laboratory study was borne out by the high eclosion rates at all diet levels compared to *An. gambiae* which was very sensitive to level. This trait also makes it a relatively insensitive choice with which to compare diets.

These results demonstrated that as far as choosing a diet, TetraMin is the least desirable for *An. gambiae* because of the sensitivity to diet level that was required for adult production; neither the highest nor lowest doses resulted in adults within what we considered a practical time period. Linenberg et al. [3] also observed that two pellet fish foods performed better than TetraMin flakes though it is not clear whether the specific product was the same as the one we tested. For genetic control mass-rearing purposes in which field performance of released mosquitoes may depend upon mating competitiveness and flight range, additional studies in large cages would be useful to ensure that the beneficial characteristics we identified in this study are correlated with these critical traits.

We were surprised that two different formulations of the Damiens diet prepared by CDC and Frontier Scientific Services gave measurably different results. There are two differences which might have contributed. Frontier used defatted liver powder whereas the CDC source did not specify whether it was defatted or not. Secondly, the Frontier team had access to a hammer mill which permitted the tuna meal to be ground more finely—likely contributing a larger amount of indigestible scale and bone to the final formulation of diet resulting in lower

**Table 7.** *Anopheles gambiae* weight statistics with significant effects shown in bold font.

|  | F | d.f. | p | R$^2$ |
|---|---|---|---|---|
| **Full model** | **11.03** | **54,731** | **<0.001** | **0.45** |
| Sex:Diet:Level |  |  | 0.91 | 0.00 |
| Sex:Diet |  |  | 0.41 | 0.00 |
| Sex:Level |  |  | 0.58 | 0.00 |
| **Minimal adequate model** | **35.78** | **15,770** | **<0.001** | **0.41** |
| **Level:Diet** | **7.93** | **8,778** | **<0.001** | **0.05** |
| **Diet (factor)** | **16.70** | **3,778** | **<0.001** | **0.09** |
| **Sex (factor)** | **80.93** | **1,778** | **<0.001** | **0.11** |
| **Feeding Level (factor)** | **909.51** | **3,778** | **<0.001** | **0.33** |

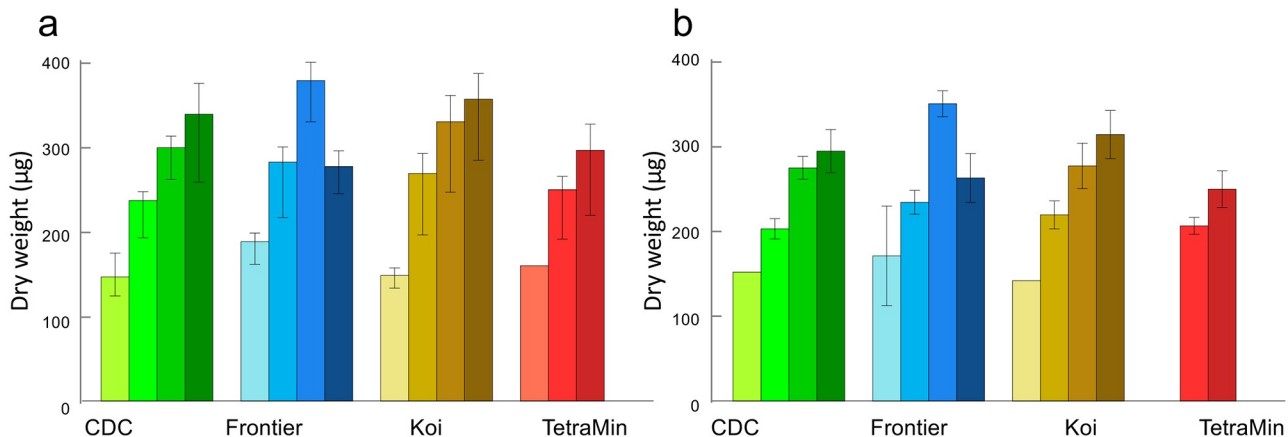

**Fig 6. *Anopheles gambiae* dry weights.** (a) *Anopheles gambiae* females and (b) males. Darker shades of color indicate increasing diet levels of 8, 16, 32 and 64 mg. Error bars are 95% confidence intervals of the mean.

concentrations of other more nutritious tuna parts. The CDC team discarded the larger particles. These two differences may have resulted in a formulation with measurably different nutritional content on a weight basis.

Of the diets tested, one can make an evaluation of their performance assuming, somewhat subjectively, that maximal survival rates, longevity and size along with short development times are desirable outcomes (Table 9).

The deviation from a 1:1 ratio of females and males that we observed in the New Orleans strain of *Ae. aegypti* is common among many strains of *Ae. aegypti* [12]. In contrast, the authors are unaware of any natural strains of *An. gambiae* that demonstrate sex ratio bias although this has been observed among progeny of crosses between different species of the *An. gambiae* complex [13].

One diet, Koi, was tested to determine whether the method of presentation of the same diet had an effect on the development rate and survival to the pupa stage. Of the other diets that could be fed in either a whole or ground form, only TetraMin is originally in a flake form and similar comparisons are possible. Any of the powders or flakes can be sprinkled on the surface, a practice which is consistent with the 'surface feeding' behavior of *Anopheles spp.* [1].

The authors are aware that some laboratories provide the diet as intact pellets or flakes rather than as a slurry. The difference between the total weights of food in our analysis confounds our analysis and arguably, if one provided more pellets, the development rates of females would be the same as when fed slurry. But as far as these analyses can be interpreted, one can conclude that for a given amount of food, increasing the immediate availability in a ground form will increase the development rate. Feeding as a slurry also allows a continuously variable (rather than discrete) amount of food to be delivered though this advantage requires mixing and pipetting slurry vs. simply counting pellets.

**Table 8. Survival and development with significant effects shown in bold font.**

| | *Ae. aegypti* | | | | | | *An. gambiae* | | | | | |
| --- | --- | --- | --- | --- | --- | --- | --- | --- | --- | --- | --- | --- |
| | Female | | | Male | | | Female | | | Male | | |
| | $\chi^2$ | d.f | p | $\chi^2$ | d.f | p | $\chi^2$ | d.f | p | $\chi^2$ | d.f | p |
| Number of pupae | 1.23 | 1 | 0.27 | 0.05 | 1 | 0.82 | 0.78 | 1 | 0.38 | 2.33 | 1 | 0.13 |
| Larval duration | **12.68** | **1** | **<0.001** | 2.43 | 1 | 0.12 | **6.5** | **1** | **<0.05** | 0.01 | 1 | 0.92 |

**Table 9. A semi-subjective assessment of the salient biological outcomes measured as an assessment of laboratory use of the four diets tested (advantageous characteristics are highlighted in green, neutral ones in gray and disadvantageous ones in yellow.).**

| | *Ae. aegypti* | | | | *An. gambiae* | | | |
|---|---|---|---|---|---|---|---|---|
| | CDC | Frontier | Koi | TetraMin | CDC | Frontier | Koi | TetraMin |
| **Survival to eclosion** | | Higher at low level | | Higher at low level | | | Highest | Lowest |
| **Probability of pupae to eclose** | | | | | | | Highest | |
| **Eclosion sensitivity to diet levels** | | Fewer adults eclosing at lowest level | | | | | Lowest | Highest |
| **Development rate** | | | | | | More consistent for all doses | | |
| **Dry weight** | | | | | | | | |
| **Adult longevity** | Higher for males | | Higher for males | | Higher for females | | Higher for females | |

Our results demonstrate that although the *An. gambiae* feeding rate in mosquito publications is often described as '*ad libitum*', it is almost certain this is never the case. The levels of diet that result in the largest size and reflect true *ad libitum* feeding activity cause so much mortality that they would not be used. Expressing it another way, larvae will continue eating more food at levels that are not consistent with overall survival of mosquitoes for experiments. In most experiments, the amount of food that is made available always restricts growth below the maximal size possible under true *ad libitum* conditions.

We consider all of the diets tested acceptable for routine laboratory purposes. However, the superior performance and low cost of the Koi food makes it a good choice for most purposes. It can be fed either as a slurry or pellet and is available in large amounts which can be frozen to stockpile the food for future use, a practice that would permit only occasional importation and ensure a long-term supply of the same batch.

## Acknowledgments

The findings and conclusions in this report are those of the authors and do not necessarily represent the official position of the Centers for Disease Control and Prevention. Use of trade names is for identification only and does not imply endorsement by the Centers for Disease Control and Prevention/the Agency for Toxic Substances and Disease Registry, the Public Health Service, or the U.S. Department of Health and Human Services. Frontier Scientific Services supplied the formulation to the CDC without charge with the understanding that the experimental design and diet comparisons would not be influenced by the potential for commercialization.

The following reagents were obtained through the NIH Biodefense and Emerging Infections Research Resources Repository, NIAID, NIH: *An. gambiae*, strain 'G3' (MRA-112) and *Ae aegypti* 'New Orleans' strain (NR-49160). The funders had no role in study design, data collection and analysis, decision to publish, or preparation of the manuscript.

## Author Contributions

**Conceptualization:** Mark Q. Benedict.

**Data curation:** Mark Q. Benedict, C. Matilda Collins.

**Formal analysis:** C. Matilda Collins.

**Funding acquisition:** Mark Q. Benedict, Ellen M. Dotson.

**Investigation:** Mark Q. Benedict, Catherine M. Hunt, Michael G. Vella, Kasandra M. Gonzalez.

**Methodology:** Mark Q. Benedict, Michael G. Vella.

**Project administration:** Mark Q. Benedict, Ellen M. Dotson.

**Resources:** Mark Q. Benedict, Michael G. Vella.

**Software:** C. Matilda Collins.

**Supervision:** Mark Q. Benedict, Ellen M. Dotson.

**Visualization:** C. Matilda Collins.

**Writing – original draft:** Mark Q. Benedict, Catherine M. Hunt, Michael G. Vella, Kasandra M. Gonzalez, Ellen M. Dotson, C. Matilda Collins.

**Writing – review & editing:** Mark Q. Benedict, Catherine M. Hunt, Michael G. Vella, Kasandra M. Gonzalez, Ellen M. Dotson, C. Matilda Collins.

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
