## [Decision Letter · Decision Letter 0]

1 Nov 2019

PONE-D-19-22729

Pragmatic selection of larval mosquito diets for insectary rearing of Anopheles gambiae and Aedes aegypti

PLOS ONE

Dear Dr. Benedict,

Thank you for submitting your manuscript to PLOS ONE. After careful consideration, we feel that it has merit but does not fully meet PLOS ONE’s publication criteria as it currently stands. Therefore, we invite you to submit a revised version of the manuscript that addresses the points raised during the review process.

We would appreciate receiving your revised manuscript by Dec 16 2019 11:59PM. To enhance the reproducibility of your results, we recommend that if applicable you deposit your laboratory protocols in protocols.io, where a protocol can be assigned its own identifier (DOI) such that it can be cited independently in the future. For instructions see: http://journals.plos.org/plosone/s/submission-guidelines#loc-laboratory-protocols

We look forward to receiving your revised manuscript.

Kind regards,

Immo A. Hansen, Ph.D.

Academic Editor

PLOS ONE

Journal Requirements:

MQB, EMD

Funded byTarget Malaria, a project that receives core funding from the Bill & Melinda Gates Foundation and from the Open Philanthropy Project Fund, an advised fund of Silicon Valley Community Foundation to the Target Malaria project, PI Austin Burt.

NO - The funders had no role in study design, data collection and analysis, decision to publish, or preparation of the manuscript.

We note that one or more of the authors are employed by a commercial company:  Frontier Scientific Services 

Reviewers' comments:

Reviewer's Responses to Questions

**Comments to the Author**

1. Is the manuscript technically sound, and do the data support the conclusions?

Reviewer #1: Yes

Reviewer #2: Partly

2. Has the statistical analysis been performed appropriately and rigorously? 

Reviewer #1: Yes

Reviewer #2: I Don't Know

3. Have the authors made all data underlying the findings in their manuscript fully available?

Reviewer #1: Yes

Reviewer #2: Yes

4. Is the manuscript presented in an intelligible fashion and written in standard English?

Reviewer #1: Yes

Reviewer #2: Yes

5. Review Comments to the Author

Reviewer #1: The study compares four different diets at four different concentrations for general colony rearing purposes, specifically using proven measures of mosquito fitness starting from larval survival and pupation to eclosion, size and survival. Importantly, two different mosquito vectors are used : one robust and one with very strict needs. The techniques are sound and the analyses appropriate to the question.

As such, I recommend publication but after a few minor changes have been made.

1. Study design in a figure format would be helpful.

2. I’m not certain why table 1 is not relegated to the supplement.

3. In Table 2, I’m assuming the column represents median survival time in days : it should be specified.

Reviewer #2: This manuscript was interesting to read and very informative to those conducting research using mosquito models. Since the manuscript mentions fatty acids, a comparison of fatty acids in each meal would be beneficial. The author also mentions fatty acids are important for flight, therefore, a flight mill test would have been a additional testing parameter for adults reared on each meal. Additional comments are below:

Line 107: Were the eggs desiccated before they were placed in water and vacuum hatched? If they were desiccated, for how long?

Line 208 and 349: For Aedes aegypti, males pupae first, so the observed “consistently male-biased” could be because the pupae that was first collected were mostly males. The author should clarify in methods the time pupation first started and the time of collections after first observing pupation. For example, if the author sees pupae on day one and collects those pupae first thing in the morning, majority of them will be males, but if the author collects pupae later that day (or even the next day), the there might be an even male to female sex ratio. The author should clarify this in the methods.

Table 9: This graph should be filled out to summarize all the data. It is awkward that only a few sections are filled.

Line 367: Remove “on” in this sentence.

6. PLOS authors have the option to publish the peer review history of their article (what does this mean?). If published, this will include your full peer review and any attached files.

Reviewer #1: No

Reviewer #2: No

---

## [Author Response · Author response to Decision Letter 0]

7 Jan 2020

Dear Editor,

Thank you for considering the manuscript titled “Pragmatic selection of larval mosquito diets for insectary rearing of Anopheles gambiae and Aedes aegypti” for publication in PLoS One. While the reviewer’s comments were not numerous, we found them extremely helpful to improve the manuscript.

In addition to our responses to the reviewer’s comments, we have made several changes, most of which were to correct typos or to clarify statements.

Our responses to the reviewer’s comments are as follows (without comments that did not request a response):

Reviewer #1:

1. Study design in a figure format would be helpful.

In response to this request we have prepared a new figure (Figure 1) which illustrates the design of the core experiments. Consequently, the other figures have been renumbered both in the text and the file names.

2. I’m not certain why table 1 is not relegated to the supplement.

Yes, that certainly could be done. We are reluctant to do so for two reasons: it is important for the readers to see those values when judging the interpretation of the conclusions and our perception is that few readers read supplemental information; secondly, we have no other materials in supplemental information. Therefore it would make a rather small amount of supplemental content.

3. In Table 2, I’m assuming the column represents median survival time in days : it should be specified.

Agreed. This was an oversight on our part. We have specified that information to the table headings.

Reviewer #2:

1. Since the manuscript mentions fatty acids, a comparison of fatty acids in each meal would be beneficial.

Yes, it would be beneficial. Unfortunately, this chemical analysis is rather specialized and we have no collaborators who are capable of performing it. However, all of the diets contain marine sources that are rich in fatty acids.

2. The author also mentions fatty acids are important for flight, therefore, a flight mill test would have been a additional testing parameter for adults reared on each meal.

The point is well taken, and we have added a caveat to our findings by adding the following statement to the Discussion section:

“For genetic control mass-rearing purposes in which field performance of released mosquitoes may depend upon mating competitiveness and flight range, additional studies in large cages would be useful to ensure that the beneficial characteristics we identified are also identified with these critical traits.

We agree that this would be useful. However, there are very few research groups who are capable of performing this, none of whom are collaborators. 

We have reservations about the value of flight-mill performance, particularly of male Anopheles which we have experience with. Males are reluctant to fly on mills, unlike females, and male data quality is rather poor.

3. Additional comments are below:

Line 107: Were the eggs desiccated before they were placed in water and vacuum hatched? If they were desiccated, for how long?

We have provided more detail in that section.

Line 208 and 349: For Aedes aegypti, males pupae first, so the observed “consistently male-biased” could be because the pupae that was first collected were mostly males.

That is not possible, but we understand why the reviewer might have concluded this. We have clarified that all pupae that formed were collected. Since the ratio is based on all male pupae, the time of pupation could have no effect. We have added language clarifying that even if all individuals that failed to develop were females, the ratio would still be male biased.

The author should clarify in methods the time pupation first started and the time of collections after first observing pupation. For example, if the author sees pupae on day one and collects those pupae first thing in the morning, majority of them will be males, but if the author collects pupae later that day (or even the next day), the there might be an even male to female sex ratio. The author should clarify this in the methods.

That is a good point. We have also added language that specifies that pupae were collected daily in the morning. So pupae were a maximum of one day old when collected.

Table 9: This graph should be filled out to summarize all the data. It is awkward that only a few sections are filled.

We agree. We have completely filled all the cells in the table according to the scheme described in the title.

Line 367: Remove “on” in this sentence.

 Thank you. We have corrected this.

---

## [Editor Report · Decision Letter 1]

7 Feb 2020

Pragmatic selection of larval mosquito diets for insectary rearing of Anopheles gambiae and Aedes aegypti

PONE-D-19-22729R1

Dear Dr. Benedict,

We are pleased to inform you that your manuscript has been judged scientifically suitable for publication and will be formally accepted for publication once it complies with all outstanding technical requirements.

With kind regards,

Immo A. Hansen, Ph.D.

Academic Editor

PLOS ONE
---

## [Editor Report · Acceptance letter]

28 Feb 2020

PONE-D-19-22729R1 

Pragmatic selection of larval mosquito diets for insectary rearing of *Anopheles gambiae* and *Aedes aegypti*

Dear Dr. Benedict:

I am pleased to inform you that your manuscript has been deemed suitable for publication in PLOS ONE. Congratulations! Your manuscript is now with our production department. 

With kind regards,

on behalf of

Dr. Immo A. Hansen 

Academic Editor

PLOS ONE